# Gastric Metastasis Mimicking Early Gastric Cancer from Invasive Ductal Carcinoma of the Breast: Case Report and Literature Review

**DOI:** 10.3390/medicina60060980

**Published:** 2024-06-13

**Authors:** Kwon Cheol Yoo, Dae Hoon Kim, Sungmin Park, HyoYung Yun, Dong Hee Ryu, Jisun Lee, Seung-Myoung Son

**Affiliations:** 1Department of Surgery, Chungbuk National University Hospital, Cheongju 28644, Republic of Korea; ykc1019@naver.com (K.C.Y.);; 2Department of Surgery, Chungbuk National University College of Medicine, Cheongju 28644, Republic of Korea; 3Department of Radiology, Chungbuk National University Hospital, Cheongju 28644, Republic of Korea; 4Department of Radiology, Chungbuk National University College of Medicine, Cheongju 28644, Republic of Korea; 5Department of Pathology, Chungbuk National University Hospital, Cheongju 28644, Republic of Korea; 6Department of Pathology, Chungbuk National University College of Medicine, Cheongju 28644, Republic of Korea

**Keywords:** invasive ductal carcinoma of breast, gastric metastasis, immunohistochemistry, cytokeratin 7

## Abstract

*Backgound and Objectives*: Gastric metastasis from invasive ductal breast cancer (BC) is rare. It mainly occurs in patients with lobular BC. The occurrence of multiple metastases is typically observed several years after the primary diagnosis. Endoscopic findings of gastric metastasis of the BC were usually the linitis plastic type. *Case presentation*: A 72-year-old women who underwent right modified radical mastectomy (MRM) 10 month ago was referred after being diagnosed with early gastric cancer (EGC) during systemic chemotherapy. EGC type I was found at gastric fundus, and pathologic finding showed poorly differentiated adenocarcinoma. Metachronous double primary tumor EGC was considered. *Management and Outcome*: A laparoscopic total gastrectomy was performed, and postoperative pathology revealed submucosa invasion and two lymph node metastases. A pathologic review that focused on immunohistochemical studies of selected antibodies such as GATA binding protein 3 (GATA3), gross cystic disease fluid protein-15 (GCDFP-15), cytokeratin 7 (CK7) was performed again, comparing previous results. As a result, gastric metastasis from BC was diagnosed. After totally laparoscopic total gastrectomy, palliative first-line chemotherapy with paclitaxel/CDDP was performed. Two months after gastrectomy, she was diagnosed with para-aortic lymph node metastasis and multiple bone metastases. She expired six months after gastrectomy. *Conclusions*: Gastric metastasis from invasive ductal carcinoma of the breast, which is clinically manifested as EGC, is a very rare condition. If there is a history of BC, careful pathological review will be required.

## 1. Introduction

Among Korean women, breast cancer (BC) is the most common cancer, and gastric cancer is the fourth most common cancer. Although the incidence of gastric cancer is lower than that of breast cancer, the cancer mortality rate is relatively high for gastric cancer [1]. BC has a relatively good prognosis compared to other aggressive cancers. However, if there is distant metastasis in BC, the prognosis is very poor. The overall 5-year survival rate for stage I, II, and III BC were 98%, 92%, and 72%, respectively, but the 5-year survival rate for BC with distant metastasis was greatly reduced to 27%. Bone metastasis (39.80%) is the most common distant metastasis of breast cancer, followed by multiple metastases (33.07%), lung metastasis (10.94%), and liver metastasis (7.34%) [2]. In relation to this, metastasis of BC to the stomach is a very rare condition, and the main histologic type of gastric metastasis from BC is lobular breast carcinoma [3]. Since the incidence of gastric cancer is the highest in Korea, metastatic cancer in the stomach from other organs tends to be overlooked. We had previously reported a case in which gastric metastasis from lobular carcinoma of breast was misdiagnosed as primary Borrmann type IV gastric cancer [4]. In the present study, we report a case of gastric metastasis from invasive ductal carcinoma of breast mimicking metachronous double primary early gastric cancer (EGC).

## 2. Case Presentation

Our study was approved by the Institutional Review Board of Chungbuk National University Hospital, Republic of Korea.

A 72-year-old female woman visited the outpatient clinic for rectal high-grade dysplasia detected during a screening test. She had a history of total thyroidectomy for thyroid cancer 25 years ago. She was admitted for endoscopic submucosal dissection for rectal cancer, and a mass in the right breast was observed during physical examination at admission. She received endoscopic submucosal dissection for rectal cancer, and ultrasonography-guided breast gun biopsy was performed during admission. On pathologic examination of the patient’s rectal cancer, the size of the tumor was 1 cm, and it was a moderately differentiated adenocarcinoma with mucosal invasion. She no longer received additional treatment for rectal cancer. During an ultrasonography examination of the right breast, a 4.9 × 4.4 × 2.2 cm irregular spiculated heterogenous echoic mass was observed. Pathologic examination revealed invasive ductal carcinoma, nuclear grade 2/3, and metastatic ductal carcinoma in the right axillary lymph node. Neoadjuvant chemotherapy, which was combined with Adriamycin, cyclophosphamide, and docetaxel, was performed. After four cycles of neoadjuvant chemotherapy, partial remission was observed. She underwent right modified radical mastectomy (MRM). Postoperative pathological examination revealed invasive ductal carcinoma, and pathologic stage ypT3(8 × 7 × 2.3 cm)ypN3(22/29). Immunohistochemical staining for estrogen receptor (ER), progesterone receptor (PR), and human growth factor receptor 2 (HER-2) were all negative (Figure 1A). After surgery, adjuvant chemotherapy based on capecitabine was performed along with radiation therapy. Nine months after MRM, gastroscopy was performed because of epigastric pain and discomfort. A gastroscopy revealed a 1 cm sized elevated mass in the posterior wall of the fundus, and endoscopic biopsy was performed (Figure 1B). Microscopic examination of the biopsy specimen showed solid nests or cords of tumor cells with histologic features resembling poorly differentiated carcinoma at initial diagnosis (Figure 1C). Immunohistochemical staining for GATA3 (GATA binding protein 3) and gross cystic disease fluid protein-15 (GCDFP-15) was carried out to rule out the possibility of metastatic breast cancer. Since both markers were negative in both breast cancer and in gastric cancer, it was inconclusive for determining the origin of the gastric tumor (Figure 2A–D). Moreover, the small size of the biopsy specimen itself brings limitations in evaluating the histologic features of entire gastric tumor. Therefore, a resection of the entire mass was needed. As the tumor cells were poorly differentiated, gastrectomy was considered more than endoscopic resection. Preoperative examination, including abdomen-pelvis CT and chest CT, was performed to check for distant metastasis, and no evidence of distant metastasis was found. She underwent totally laparoscopic total gastrectomy 10 months after MRM. The patient was diagnosed with T1bN1M0 gastric cancer according to AJCC 8th edition. Gross examination of the resected specimen revealed a well-demarcated mass measuring 1 × 0.7 cm. The entire mass was microscopically examined, and a poorly differentiated carcinoma that invaded the submucosa was identified; however, it differed from the general histological features of primary gastric adenocarcinoma for the following reasons: (1) No surface epithelial cells showed dysplasia or malignant change. (2) All tumor cells were located in the lamina propria with a solid nest or cord patterns with no glandular growth pattern (Figure 3A). Therefore, the tumor was more likely to be metastatic breast cancer than primary gastric cancer. It was reviewed again with the pathological findings of previous BC, and the histologic features of breast cancer and gastric cancer were almost identical. Additionally, cytokeratin CK 7 (CK7) immunostaining showed diffuse and strong positivity in both the gastric and breast specimens (Figure 3B,C). As a result, possible gastric metastasis of BC was diagnosed because the histological shape and immunohistochemical staining findings were consistent with BC. After the totally laparoscopic total gastrectomy, the patient received palliative first-line chemotherapy with paclitaxel and cisplatin (CDDP). Two months after gastrectomy, she was diagnosed with para-aortic lymph node metastasis and multiple bone metastases (Figure 4A,B). She underwent hip arthroplasty due to pathologic fracture of the right hip joint three months after gastrectomy, and she expired six months after gastrectomy.

## 3. Discussion

Gastric cancer is the most common malignancy in Korea. In Korean women, BC is the most common cancer, and gastric cancer the fourth most common cancer [1]. Colorectal and gastric cancers are known to share common risk factors, such as dietary habits and genetic predispositions. The rate of metachronous double primary cancer diagnosed after detection of gastric cancer in Korea was 3.7–4.8%, with the most common metachronous double primary being colon cancer [5,6]. The relationships between other cancers like thyroid and breast cancer are less clear. Thyroid cancer has been associated with radiation exposure and certain genetic syndromes, while breast cancer is often linked to hormonal and genetic factors. Breast cancer (BC) is the most common malignancy in women, and is associated with a considerable risk of developing multiple primary cancers (MPCs) due to factors such as increased patient survival, genetic susceptibility, and environmental interactions. The incidence of a secondary primary cancer after BC is between 4% and 16%, with the most common types being thyroid cancer and gynecological malignancies. The mammary glands are closely related to the female reproductive system, which explains why gynecologic malignancies are the most common primary cancers following BC [7]. 

In our case, the patient had a history of thyroid cancer and rectal cancer, which might suggest a potential association with breast cancer. However, gastric metastasis from breast cancer is rare and thus often overlooked, especially in regions like Korea where gastric cancer is prevalent. The incidence of metachronous double primary breast cancer (BC) in Korean women diagnosed with gastric cancer is higher than that of metachronous double primary colorectal cancer. Therefore, metastasis to the stomach from BC tends to be overlooked because metachronous or synchronous double primary BC is relatively common after a diagnosis of gastric cancer. Additionally, it is noteworthy that double primary malignancies, especially thyroid cancer, are the most common secondary malignancies following BC [7], which might lead to the oversight of metastatic gastric cancer in regions like Korea where gastric cancer is prevalent.

We previously reported metastatic gastric cancer from invasive lobular carcinoma of the breast. In that case, the patient who underwent MRM because of invasive lobular carcinoma of the breast complained of epigastric discomfort during the hospitalization period. So, she received an endoscopy during the hospitalization period, and the endoscopy revealed Borrmann type IV gastric cancer. Advanced gastric cancer was considered as a synchronous double primary cancer. However, on pathological review, gastric metastasis from invasive lobular carcinoma of the breast was diagnosed. Immunohistochemical staining demonstrated positivity of ER, PR, and GCDFP-15; however, cytokeratin 5 (CK5) was negative in primary BC. Additionally, the same immunohistochemical staining result was shown in the gastric lesion. Based on these immunohistochemical exams, gastric metastasis from breast cancer was diagnosed and unnecessary gastrectomy was prevented. Immunohistochemical analysis was helpful in diagnosing gastric metastases of BC [4]. 

In this case, an endoscopy was performed nine months after MRM, and it showed EGC. The possibility of gastric metastasis of BC has been overlooked for several reasons. First, the incidence of gastric cancer is very high in Korea, but the gastric metastasis of BC is a very rare condition. Additionally, Gastric metastasis from breast cancer mainly occurred in invasive lobular carcinoma of the breast [8,9]. In the autopsy study, the rates of gastric metastasis from BC were 4~10% [10,11,12]. Gastric metastasis of BC may occur many years after the diagnosis of BC, and 90–94% of patients with gastric metastasis have concurrent other distant metastasis [3,8]. In our case, gastric metastasis of BC was detected relatively quickly nine months later, and there was no evidence of other concurrent distant metastasis on abdominal and pelvic CT that was performed before the gastrectomy. Additionally, its pathologic finding was invasive ductal carcinoma that had less incidence of gastric metastasis than invasive lobular carcinoma. In addition, the rate of metachronous double primary tumor after gastric cancer diagnosis is not low in Korea [5], and physicians were biased due to the patient’s history of thyroid cancer and rectal cancer. These findings led to a misdiagnosis of metachronous double primary gastric cancer rather than gastric metastasis of BC.

Second, in the patient’s primary BC lesion, ER, PR, HER-2, and CK5 were all negative. Additional immunohistochemical studies for GATA3 and GCDFP-15 were performed for differentiation between double primary gastric cancer and gastric metastasis; the results were all negative. Because of these results, it was confusing to diagnose gastric metastasis of BC. Immunohistochemical studies aid in the analysis of distant metastasis, particularly in the absence of a definitive history of BC, or when there is a history of BC [13]. GATA3 is part of the zinc finger transcription factor family, and it plays a very important role in the differentiation of many tissues such as breast glandular epithelial cells, hair follicles, T lymphocytes, adipose tissue, kidney, and the nervous system [14,15]. The sensitivity of GATA3 is much higher in invasive lobular carcinoma than invasive ductal carcinoma of the breast [16,17]. Liu et al. reported that GATA3 was expressed at a lower level in ER-negative BC, showing a sensitivity of 69%, and the sensitivity was significantly lower in triple-negative breast cancer (TNBC was defined when ER, PR, and HER-2 are all negative). GCDFP-15 is a 15 kDa protein originally detected in cystic fluid from cystic mastopathy [18]. It is not expressed in normal ductal or lobular epithelium, but expressed in the apocrine metaplasia of the breast [19]. Wick et al. [20] reported that besides BC, the tumors that expressed GCDFP-15 were carcinoma of the salivary gland, sweat gland, and prostate, since three types of lesions are unlikely to be diagnosed as metastatic BC. After excluding them, the adjusted specificity of GCDFP-15 and the predictive value of a positive result for metastatic diagnosis of BC were both 99%. Therefore, it is highly specific for female mammary gland differentiation, and is frequently used as an immunohistochemical marker for the evaluation of potential mammary gland origins of metastatic carcinoma of unknown primary site [21]. In addition to their specific histological features, GCDFP-15 and GATA3 have low sensitivity in triple-negative BC; thus, triple-negative breast cancer at the metastasis site is difficult to diagnose [13]. Unfortunately, our case presented triple breast cancer, and GATA3 and GCDFP-15 immunochemical staining were negative. These immunohistochemical staining findings were the cause of misdiagnosis of metastatic breast cancer as EGC.

Third, the endoscopic findings of gastric metastasis from breast cancer are mostly the linitis plastica type, which infiltrates the submucosa and muscularis propria via diffusion; less commonly, discrete nodular or external compression may occur [3,8,22,23]. In our case, the endoscopic finding and clinical features were EGC. The endoscopic morphology was EGC type I, according to the Paris classification of superficial neoplastic lesions in the digestive tract [24]. Gastric metastases in breast cancer are mostly diagnosed as advanced gastric cancer, and in most cases, are accompanied by distant metastases to other organs [3,8]. 

## 4. Conclusions

In summary, the diagnosis was invasive ductal carcinoma; the immunohistochemical studies presented triple-negative breast cancer, and its endoscopic morphology was EGC. For these reasons, gastric metastasis of breast cancer was misdiagnosed as metachronous primary EGC. The pathological examination performed on the specimen after gastrectomy did not show general histological features of primary gastric cancer, so we reviewed the findings again along with the pathologic findings of previous breast cancer diagnoses with the addition of CK7. Triple-negative breast cancer does not have a standard immune profile, and it can be difficult to differentiate between primary and metastatic lesions because site-specific markers such as GATA3, GCDFP-15, and mammaglobin are often missing [25]. Stats et al. [26] reported that a panel of CK7, GATA3, and Sry-related HMg-Box gene 10 (SOX-10) is complementary in the diagnosis of brain metastasis in breast cancer. Although there are no studies for CK7 in cases of gastric metastasis of primary breast cancer, Chu et al. reported that CK7 has a positive rate of 96% and 38% in breast cancer and gastric cancer, respectively [27]. Since the expression frequency of CK7 is relatively lower in gastric cancer than in breast cancer, CK7 may thus be helpful in the diagnosis of gastric metastases of breast cancer. In addition, although it is a very specific case, endoscopic submucosal resection is considered to be helpful in pathological diagnosis for gastric metastasis of breast cancer that EGC in endoscopic findings, as in this case.

In primary early gastric cancer, the standard treatment approach is surgical resection. However, in cases where there is a history of advanced breast cancer, it is crucial to perform immunohistochemical examinations to differentiate between a second primary cancer and metastatic disease. This distinction is vital because metastasectomy for breast cancer metastases has not been shown to significantly improve survival rates. Studies indicate that while metastasectomy might offer some survival benefit in highly selected patients with solitary metastases and favorable prognostic factors, it does not generally confer a significant survival advantage over systemic therapy alone for most patients [28]. Therefore, if the gastric lesion is identified as metastatic breast cancer, systemic chemotherapy should be considered instead of surgical resection. According to the National Comprehensive Cancer Network (NCCN) guidelines, the treatment for metastatic breast cancer (MBC) is primarily systemic chemotherapy, as it is the mainstay of treatment for managing metastatic disease. Surgical resection in this context is generally limited to improving quality of life or providing symptomatic relief rather than enhancing survival [29]. This approach underscores the importance of accurate pathological diagnosis to guide appropriate treatment strategies. For patients with a history of advanced breast cancer presenting with a gastric lesion that mimics early gastric cancer, thorough immunohistochemical analysis is necessary to ensure the correct diagnosis and treatment plan. It is very rare for invasive ductal carcinoma of the breast to have gastric metastasis, and it is very rare for endoscopic findings to show EGC. Triple-negative breast cancer is also very difficult to diagnose between metastatic tumors and primary tumors. If there is a history of advanced breast cancer, even if it shows the form of early gastric cancer, gastric metastasis of breast cancer should be considered, and careful pathological diagnosis is required. In addition, CK7 and endoscopic submucosal resection are thought to be helpful in differentiating gastric metastases in breast cancer from primary gastric cancer.

## Figures and Tables

**Figure 1 medicina-60-00980-f001:**
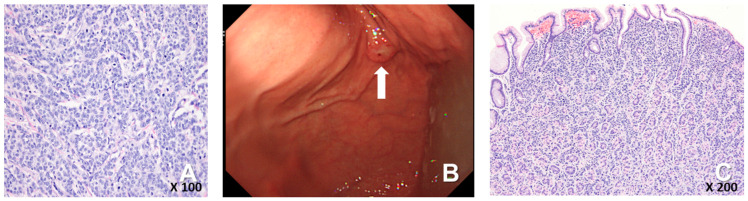
(**A**) The breast cancer consisted of solid nests or cords of tumor cells, which was consistent with the diagnosis of invasive ductal carcinoma (H-E staining, ×200). (**B**) Endoscopy showed about 1 cm sized elevated out in posterior wall of gastric fundus, (white arrow). (**C**) The biopsy specimen of the gastric tumor showed poorly differentiated carcinoma with no glandular differentiation (H-E staining, ×100).

**Figure 2 medicina-60-00980-f002:**
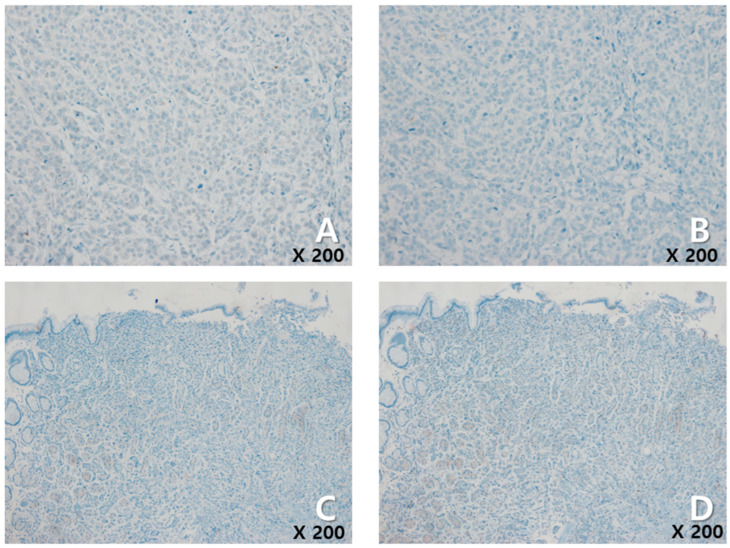
Immunohistochemical staining for GATA3 (**A**) and GCDFP-15 (**B**) was negative in the breast cancer specimen (×200). GATA3 (**C**) and GCDFP-15 (**D**) immunostaining were also negative in the biopsy specimen of the gastric tumor (×200).

**Figure 3 medicina-60-00980-f003:**
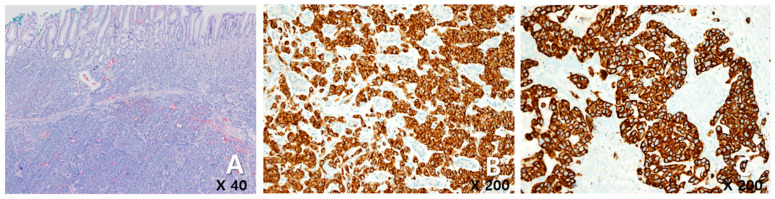
(**A**) The gastrectomy specimen showed tumor cells with histopathologic features identical to those of carcinoma of the breast, which invaded the submucosa of the stomach (×40). The cytokeratin 7 immunostaining showed diffuse and strong positivity in both gastric cancer (**B**) and breast cancer (**C**).

**Figure 4 medicina-60-00980-f004:**
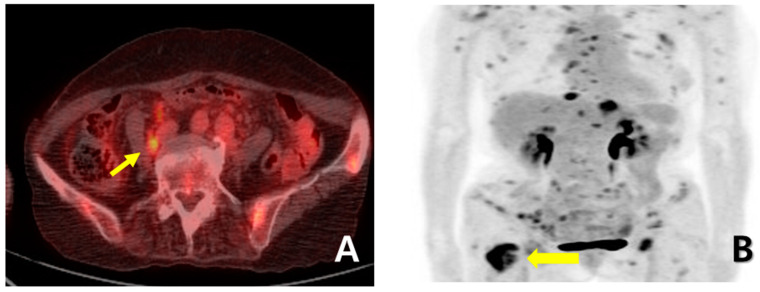
(**A**) PET-CT performed two months after total gastrectomy showed para-aortic lymph node metastasis (yellow arrow); (**B**) PET-CT also showed multiple bone metastases (yellow arrow).

## Data Availability

The original contributions presented in the study are included in the article. Further inquiries can be directed to the corresponding author.

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
