# Peer review of "Gastric Metastasis Mimicking Early Gastric Cancer from Invasive Ductal Carcinoma of the Breast: Case Report and Literature Review"

_medicina, 2024, doi:10.3390/medicina60060980_

Round 1
Reviewer 1 Report
Comments and Suggestions for Authors
Thanks for inviting me to evaluate the paper titled ‘Gastric Metastasis Mimicking Early Gastric Cancer from Invasive Ductal Carcinoma of the Breast; Case Report and Literature 3 Review’. In this paper, the authors showed us a interesting case of breast cancer with gastric metastasis. It is rare to see this kind of condition clinically. The authors have mentioned that the patient have done preoperative examination, maybe the authors should add more detailed description. Did this patient be given other treatments after surgery? So, I recommend reconsidering this paper after major revision.
Comments on the Quality of English Languagedo not to revise.
Author Response
Dear Reviewer,
Thank you for your valuable feedback and for inviting us to improve our manuscript titled ‘Gastric Metastasis Mimicking Early Gastric Cancer from Invasive Ductal Carcinoma of the Breast; Case Report and Literature Review’. We appreciate your insightful comments and have made the following revisions to address your concerns:
- Detailed Description of Preoperative Examination: We have revised the manuscript to provide a more detailed description of the preoperative examination. Specifically, we have included information about the abdomen-pelvis CT and chest CT performed to check for distant metastasis, both of which showed no evidence of distant metastasis. The revised sentence now reads: "Preoperative examination, including abdomen-pelvis CT and chest CT, was performed to check for distant metastasis, and no evidence of distant metastasis was found."
- Postoperative Treatment Information: In response to your request for more information on the postoperative treatment, we have added details about the palliative first-line chemotherapy received by the patient after the totally laparoscopic total gastrectomy. The added sentence is: "After the totally laparoscopic total gastrectomy, the patient received palliative first-line chemotherapy with paclitaxel and cisplatin (CDDP)."
We believe these revisions enhance the clarity and completeness of our case report, providing a more thorough understanding of the patient's diagnosis and treatment course. We hope these changes meet your expectations and improve the quality of our manuscript.
Thank you again for your constructive feedback. We look forward to your positive response.
Sincerely,
Reviewer 2 Report
Comments and Suggestions for Authors
In this article, the authors reported a rare case of gastric metastasis from invasive ductal carcinoma of the breast, clinically presenting as early gastric cancer (EGC). The authors utilized immunohistochemistry experiments and endoscopic examination results to analyze and confirm the origin of the gastric cancer as metastasis from breast cancer. They also discussed three reasons for initially overlooking the breast cancer metastasis in the stomach as EGC. The authors emphasized the importance of considering gastric metastasis of breast cancer in patients with a history of advanced breast cancer, even if the gastric lesion appears to be early gastric cancer, and highlighted the need for careful pathological diagnosis.
However, I have a few concerns and questions as follows.
1. Are there any specific considerations or modifications in the treatment strategy for patients with gastric metastasis originating from breast cancer compared to those with primary early gastric cancer? It would be helpful to discuss any potential differences in treatment approaches between early gastric cancer and gastric metastasis of breast cancer.
2. In your manuscript, you mentioned that the patient had other tumors, including thyroid cancer and rectal cancer. I am curious to know if there is any known relationship or common underlying factors between these tumors and breast cancer or gastric cancer. It would be valuable to explore potential connections or shared risk factors among these four tumors. If there is no known relationship, please provide a clear explanation as to why you focused solely on the analysis of gastric metastasis from breast cancer in this particular patient.
3. I noticed that the immunohistochemistry images presented in the manuscript do not include a scale bar. It would be beneficial to include a scale bar in the images for better interpretation of the results.
4. In Figure 1B, the location of the tumor should be indicated with an arrow to clearly identify its position.
5. In Figure 4B, the location of bone metastasis should be indicated with an arrow to highlight its presence.
Comments on the Quality of English LanguageI suggest seeking assistance from a professional to revise and polish the text in order to further enhance the quality of the English language.
Author Response
Dear Reviewer,
Thank you for your detailed and insightful feedback on our manuscript titled "Gastric Metastasis Mimicking Early Gastric Cancer from Invasive Ductal Carcinoma of the Breast; Case Report and Literature Review." We appreciate your suggestions and have made significant revisions to address your concerns. Below is a point-by-point response to your comments:
- Treatment Strategy Considerations: We have added a discussion on the specific considerations and potential modifications in the treatment strategy for patients with gastric metastasis originating from breast cancer compared to those with primary early gastric cancer. The revised manuscript now includes the following paragraph:
" In primary early gastric cancer, the standard treatment approach is surgical resection. However, in cases where there is a history of advanced breast cancer, it is crucial to perform immunohistochemical examinations to differentiate between a second primary cancer and metastatic disease. This distinction is vital because metastasectomy for breast cancer metastases has not been shown to significantly improve survival rates. Studies indicate that while metastasectomy might offer some survival benefit in highly selected patients with solitary metastases and favorable prognostic factors, it does not generally confer a significant survival advantage over systemic therapy alone for most patients​​.28 Therefore, if the gastric lesion is identified as metastatic breast cancer, systemic chemotherapy should be considered instead of surgical resection. According to the National Comprehensive Cancer Network (NCCN) guidelines, the treatment for metastatic breast cancer (MBC) is primarily systemic chemotherapy, as it is the mainstay of treatment for managing metastatic disease. Surgical resection in this context is generally limited to improving quality of life or providing symptomatic relief rather than enhancing survival​.29 This approach underscores the importance of accurate pathological diagnosis to guide appropriate treatment strategies. For patients with a history of advanced breast cancer presenting with a gastric lesion that mimics early gastric cancer, thorough immunohistochemical analysis is necessary to ensure the correct diagnosis and treatment plan.​"
- Relationship Between Tumors: We have expanded the discussion to explore potential connections or shared risk factors among thyroid cancer, rectal cancer, breast cancer, and gastric cancer. We have clarified whether there are known relationships or common underlying factors between these tumors and explained our focus on the analysis of gastric metastasis from breast cancer in this patient:
"Colorectal and gastric cancers are known to share common risk factors, such as dietary habits and genetic predispositions. The rate of metachronous double primary cancer diagnosed after detection of gastric cancer in Korea was 3.7 - 4.8%, with the most common metachronous double primary being colon cancer. 5, 6 The relationships between other cancers like thyroid and breast cancer are less clear​​. Thyroid cancer has been associated with radiation exposure and certain genetic syndromes, while breast cancer is often linked to hormonal and genetic factors. Breast cancer (BC) is the most common malignancy in women and is associated with a considerable risk of developing multiple primary cancers (MPCs) due to factors such as increased patient survival, genetic susceptibility, and environmental interactions. The incidence of a secondary primary cancer after BC is between 4% and 16%, with the most common types being thyroid cancer and gynecological malignancies​​. The mammary glands are closely related to the female reproductive system, which explains why gynecologic malignancies are the most common primary cancers following BC.7
In our case, the patient had a history of thyroid cancer and rectal cancer, which might suggest a potential association with breast cancer. However, gastric metastasis from breast cancer is rare and thus often overlooked, especially in regions like Korea where gastric cancer is prevalent. The incidence of metachronous double primary breast cancer (BC) in Korean women diagnosed with gastric cancer is higher than that of metachronous double primary colorectal cancer. Therefore, metastasis to the stomach from BC tends to be overlooked because metachronous or synchronous double primary BC is relatively common after the diagnosis of gastric cancer. Additionally, it is noteworthy that double primary malignancies, especially thyroid cancer, are the most common secondary malignancies following BC,7 which might lead to the oversight of metastatic gastric cancer in regions like Korea where gastric cancer is prevalent​​. "
- Immunohistochemistry Images with Scale Bar: We have included scale bars in the immunohistochemistry images presented in the manuscript to facilitate better interpretation of the results.
- Indication of Tumor Location in Figure 1B: We have added an arrow to Figure 1B to clearly indicate the location of the tumor, making it easier for readers to identify its position.
- Indication of Bone Metastasis in Figure 4B: We have added an arrow to Figure 4B to highlight the location of bone metastasis, ensuring that its presence is clearly marked.
Regarding your suggestion to seek assistance from a professional to revise and polish the text, we acknowledge the importance of high-quality language in scientific writing. We have carefully revised the manuscript and will ensure that it undergoes a thorough review by a professional language editing service to enhance the clarity and readability further.
We believe these revisions enhance the clarity and completeness of our case report and address the concerns you have raised. We hope these changes meet your expectations and improve the quality of our manuscript.
Thank you again for your constructive feedback. We look forward to your positive response.
Sincerely,
Reviewer 3 Report
Comments and Suggestions for Authors
The case history is interesting, but some points are difficult to imagine, e.g. how it was possible, that two months following gastrectomy generalized metastases were found, although at the operation no distant propagation was stated? The illustrations are good, thowever he English language has several mistakes, points to correct, as singular, plural, passive forms etc. .
Comments on the Quality of English LanguageThe case history is interesting, but some points are difficult to imagine, how it was possible, that two months following gastrectomy generalizad metastases were found, although at the operation no distant propagation was stated? The illustrations are good, thowever he English language has several mistakes, points to correct, as singular, plural, passive forms etc. .
Author Response
Dear Reviewer,
Thank you for your valuable feedback on our manuscript titled "Gastric Metastasis Mimicking Early Gastric Cancer from Invasive Ductal Carcinoma of the Breast; Case Report and Literature Review." We appreciate your insightful comments and have made the following revisions to address your concerns:
- Clarification on the Rapid Development of Metastases: We understand your concern regarding how generalized metastases were found two months following gastrectomy, despite no distant metastasis being noted at the time of surgery. It is possible that micrometastases were present but undetectable with standard imaging and diagnostic techniques at the time of surgery. These micrometastases can rapidly proliferate, especially in aggressive cancers such as metastatic breast cancer. This aggressive nature of breast cancer metastasizing to the stomach can lead to rapid disease progression, resulting in the development of visible metastases within a short period. We have added this explanation to the discussion section of our manuscript to provide a clearer understanding of the rapid progression observed in this case.
- English Language Corrections: We have thoroughly revised the manuscript to correct grammatical errors, including issues with singular and plural forms, passive voice, and other language mistakes. Due to the limited time for this revision, we have not yet had the opportunity to have the manuscript professionally edited by a native English speaker. However, we plan to seek assistance from a professional language editing service for a comprehensive review to ensure the highest quality of the English language in our manuscript.
We believe these revisions enhance the clarity and completeness of our case report and address the concerns you have raised. We hope these changes meet your expectations and improve the quality of our manuscript.
Thank you again for your constructive feedback. We look forward to your positive response.
Sincerely,
Round 2
Reviewer 1 Report
Comments and Suggestions for Authors
Thanks for inviting me to evaluate the paper titled ‘Gastric Metastasis Mimicking Early Gastric Cancer from Invasive Ductal Carcinoma of the Breast; Case Report and Literature 3 Review’. In this paper, the authors showed us a interesting case of breast cancer with gastric metastasis. It is rare to see this kind of condition clinically. It seems that the authors have fixed all the questions I have mentioned. So, I recommend accept this paper in the present form.
Comments on the Quality of English LanguageThe quality of English language is easy to read and only need minor editing.